# Dementia risk and dynamic response to exercise: A non-randomized clinical trial

Eric D. Vidoni[1]*, Jill K. Morris[1], Jacqueline A. Palmer[2], Yanming Li[3], Dreu White[2], Paul J. Kueck[1], Casey S. John[1], Robyn A. Honea[1], Rebecca J. Lepping[1], Phil Lee[4], Jonathan D. Mahnken[3], Laura E. Martin[5], Sandra A. Billinger[1]

1 Department of Neurology, University of Kansas Medical Center, Kansas City, KS, United States of America,
2 Department of Physical Therapy, Rehabilitation Science and Athletic Training, University of Kansas Medical Center, Kansas City, KS, United States of America, 3 Department of Biostatistics and Data Science, University of Kansas Medical Center, Kansas City, KS, United States of America, 4 Department of Radiology, University of Kansas Medical Center, Kansas City, KS, United States of America, 5 Department of Population Health, University of Kansas Medical Center, Kansas City, KS, United States of America

* evidoni@kumc.edu

## Abstract

### Background

Physical exercise may support brain health and cognition over the course of typical aging. The goal of this nonrandomized clinical trial was to examine the effect of an acute bout of aerobic exercise on brain blood flow and blood neurotrophic factors associated with exercise response and brain function in older adults with and without possession of the Apolipoprotein epsilon 4 (APOE4) allele, a genetic risk factor for developing Alzheimer's. We hypothesized that older adult APOE4 carriers would have lower cerebral blood flow regulation and would demonstrate blunted neurotrophic response to exercise compared to noncarriers.

### Methods

Sixty-two older adults (73±5 years old, 41 female [67%]) consented to this prospectively enrolling clinical trial, utilizing a single arm, single visit, experimental design, with post-hoc assessment of difference in outcomes based on APOE4 carriership. All participants completed a single 15-minute bout of moderate-intensity aerobic exercise. The primary outcome measure was change in cortical gray matter cerebral blood flow in cortical gray matter measured by magnetic resonance imaging (MRI) arterial spin labeling (ASL), defined as the total perfusion (area under the curve, AUC) following exercise. Secondary outcomes were changes in blood neurotrophin concentrations of insulin-like growth factor-1 (IGF-1), vascular endothelial growth factor (VEGF), and brain derived neurotrophic factor (BDNF).

### Results

Genotyping failed in one individual (n = 23 APOE4 carriers and n = 38 APOE4 non-carriers) and two participants could not complete primary outcome testing. Cerebral blood flow AUC increased immediately following exercise, regardless of APOE4 carrier status. In an exploratory regional analyses, we found that cerebral blood flow increased in hippocampal brain

**Data Availability Statement:** The data for this manuscript can be found in the Harvard Dataverse at: https://doi.org/10.7910/DVN/YCO2SE.

**Funding:** This study was funded by grants from the National Institutes of Health R21 AG061548 (EDV, SAB, LM, PL, JDM, JKM), P30 AG072973 (N/A) and P30 AG035982 (N/A), and the Leo and Anne Albert Charitable Trust.(EDV) The Hoglund Biomedical Imaging Center imaging equipment is supported by a generous gift from Forrest and Sally Hoglund (N/A), and funding from the National Institutes of Health including S10 RR29577 (N/A) and UL1 TR002366 (N/A). The funders had no role in study design, data collection and analysis, decision to publish, or preparation of the manuscript. The content is solely the responsibility of the authors and does not necessarily represent the official views of the National Institutes of Health or other funding organizations.

**Competing interests:** The authors have declared that no competing interests exist.

regions, while showing no change in cerebellum across both groups. Among high inter-individual variability, there were no significant changes in any of the 3 neurotrophic factors for either group immediately following exercise.

## Conclusions

Our findings show that both APOE4 carriers and non-carriers show similar effects of exercise-induced increases in cerebral blood flow and neurotrophic response to acute aerobic exercise. Our results provide further evidence that acute exercise-induced increases in cerebral blood flow may be regional specific, and that exercise-induced neurotrophin release may show a differential effect in the aging cardiovascular system. Results from this study provide an initial characterization of the acute brain blood flow and neurotrophin responses to a bout of exercise in older adults with and without this known risk allele for cardiovascular disease and Alzheimer's disease.

## Trial registration

Dementia Risk and Dynamic Response to Exercise (DYNAMIC); Identifier: NCT04009629.

## Introduction

Many diseases of the brain and cardiovascular system share common risk factors such as hypertension, hypercholesterolemia, and genetics [1–4]. High comorbidity of cognitive decline and cardiovascular disease has focused much research on the role of cardio- and cerebrovascular health in reducing dementia risk [5, 6]. Aerobic exercise–characterized as sustained, rhythmic physical activity using large muscle groups—is a well-known cardiovascular intervention [7] that shows positive effects on brain health [8], including improved cognitive outcomes [9–13], greater brain volume and cortical thickness [14–16], and lower risk of dementia [17, 18]. Randomized control trials (RCTs) involving aerobic exercise have consistently demonstrated benefits to cognition and structural brain integrity, including increased volume of the whole brain and the hippocampus, a critical neural substrate for memory formation and retention that is commonly compromised with aging [9, 12, 19–21]. Increased cerebral blood flow (CBF) and exposure to blood-based trophic and hormonal factors may be key factors amongst many potential mechanisms by which aerobic exercise exerts neuroprotective and therapeutic effects on brain health. While the positive effects of long-term aerobic exercise on brain health are well established, the acute response of CBF and blood-based neurotrophins to a bout of aerobic exercise by older adults remains insufficiently characterized [22–25].

Possession of the Apolipoprotein E allele 4 (APOE4), the strongest known genetic risk factor for sporaidc Alzheimer's disease (AD), may influence the relationship between cardiovascular and brain health. Apolipoprotein E plays an integral role in maintenance of cerebrovascular health and may interact with aging processes to mediate benefits of long-term exercise interventions on brain health [26, 27]. Individuals who carry the APOE4 isoform demonstrate poor brain vascular function with aging compared to their age-matched counterparts [2–4, 28–30], particularly in regions associated with Alzheimer's disease (AD) [31, 32]. Yet, whether older individuals who possess APOE4 show differential CBF responses to an acute bout of exercise has not been investigated. Understanding acute physiologic responses to is important as any benefits of exercise will necessarily result from the cumulative effects of these brief regular exercise exposures.

To further characterize potential intermediary mechanisms between exercise and brain health we designed the present study to assess the immediate CBF response to a single bout of acute exercise. Assessing acute exercise has the benefit of providing information on the immediate changes are related to component parts of a habitual exercise program. Our driving premise was that CBF would be a biomarker of cerebrovascular change. Specifically, we hypothesized that APOE4 carriers would have lower CBF response immediately following exercise. As ancillary outcomes we also assessed vascular endothelial growth factor (VEGF), insulin-like growth factor 1 (IGF1), and brain derived neurotrophic factor (BDNF), since they have been postulated as possible neuroprotective and therapeutic mediators of exercise effects on the brain [21, 33].

## Methods

The study was designed as a single arm, single visit, experimental study, with post-hoc assessment of difference based on APOE4 carriership. No randomization was used in this study. The protocol was approved by the University of Kansas Medical Center Institutional Review Board. All participants provided written informed consent consistent with the Declaration of Helsinki. This study was registered as a clinical trial (ClinicalTrials.gov, NCT04009629) following National Institutes of Health guidance. Additional information for study design and reporting may be found in S1 Protocol and S1 Checklist.

Sixty-two English speaking adults, aged 65–85, were enrolled in the study between October 25, 2019 and October 28, 2021. To our knowledge, there have been no peer-reviewed reports of genotype-based CBF differences in response to acute exercise. Thus, we conservatively chose an estimated effect size ($d = 0.85$) based on feasibility and prior cross-sectional data [31, 34]. We calculated that enrolling a total of 60 participants would provide ~90% power with a Type 1 error rate of 5% to detect APOE4-related differences in CBF.

Exclusion criteria were musculoskeletal or cardiopulmonary restrictions from a physician; contraindications to MRI; anti-coagulant use; previous diagnosis of a cognitive disorder or a neurological or psychiatric condition that could result in cognitive impairment; high exercise risk classification by American College of Sports Medicine criteria unless cleared by a physician. Fig 1 provides a CONSORT-style diagram of enrollment. All testing was performed at the University of Kanas Medical Center. Participants were compensated $100 for their time.

We have previously described our protocol for the present study and detailed method for measuring CBF before and after a single, 15-minute acute bout of moderate intensity aerobic exercise on a cycle ergometer. Intensity was titrated to 45–55% of heart rate reserve, based on age-predicted heart rate maximum [35]. The full trial protocol is described in White et al. [35].

The primary outcome was cortical gray matter cerebral blood flow (CBF) response, quantified by area under the curve post-exercise. Neurotrophic factor concentration change from pre- to post-exercise and regional CBF response were identified as an ancillary outcome of interest *a priori*. There were no changes to trial primary outcome after the trial commenced.

Arterial spin labeling via magnetic resonance imaging (MRI) was chosen to capture CBF due to its advantage in spatial localization and ability to yield a physiologically quantifiable outcome [36]. For CBF measurement, participant underwent two 3D GRASE pseudo-continuous arterial spin labeling (pCASL) sequences [37–40], yielding 11 minutes and 36 seconds of pre-exercise CBF data. All pCASL sequences were collected with the same with background suppressed 3D GRASE protocol (TE/TR = 22.4/4300 ms, FOV = $300 \times 300 \times 120$ mm$^3$, matrix = $96 \times 66 \times 48$, Post-labeling delay = 2s, 4-segmented acquisition without partial Fourier transform reconstruction, readout duration = 23.1 ms, total scan time 5:48, 2 M0 images). The two pre-exercise pCASL sequences were followed by a T1-weighted, 3D magnetization

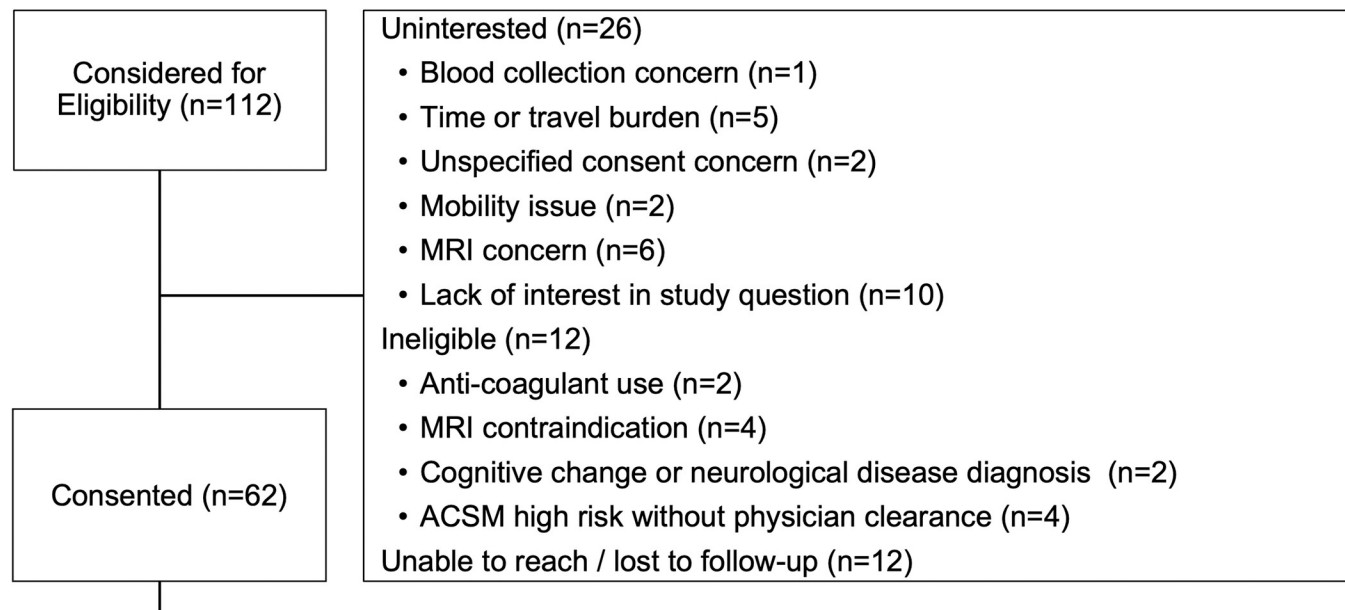

**Fig 1. CONSORT enrollment flow.**

prepared rapid gradient echo (MPRAGE) structural scan (TR/TE = 2300/2.95 ms, inversion time (TI) = 900 ms, flip angle = 9 deg, FOV = 253 × 270 mm, matrix = 240 × 256 voxels, voxel in-plane resolution = 1.05 × 1.05 mm2, slice thickness = 1.2 mm, 176 sagittal slices, in-plane acceleration factor = 2, acquisition time = 5:09). Blood pressure was monitored during the MRI via a continuous blood pressure monitoring cuff (Caretaker 4, Caretaker Medical N. A. caretakermedical.net).

Following the MRI, in an adjacent room, a flexible intravenous catheter was placed, and 10 mL of blood was collected in tubes containing ethylenediaminetetraacetic acid, and a separate 3mL sample in acid citrate dextrose for genotyping. Participants then sat on a cycle ergometer and, after a 5-minute warm-up, exercised for 15 minutes at a moderate intensity, 45–55% of heart rate reserve, on a cycle ergometer. Cycle resistance was titrated for the entire 15 minutes to maintain intensity. During a cooldown period, an additional 10mL of blood was drawn. Then participants were then escorted back to the MRI immediately for 4 consecutive pCASL sequences, identical to the pre-exercise sequences. Finally, an additional 10mL of blood was drawn.

### Neuroimage processing

CBF was calculated using a process adapted from the Laboratory of Functional MRI Technology CBF Preprocess and Quantify packages for CBF calculation (loft-lab.org, ver. February

2019). We created individualized gray matter regions of interest (whole brain, hippocampus, and cerebellum as a reference region) for each participant using the Statistical Parametric Mapping CAT12 (neuro.uni-jena.de/cat, r1059 2016-10-28) package for anatomical segmentation [41]. We motion corrected labeled and control pCASL images separately for each sequence, realigning each image to the first peer image following M0 image acquisition. CBF was calculated with surround subtraction of each label/control pair without biopolar gradients [42] producing a timeseries of 9 subtraction images. This was done for each pCASL sequence, or 18 pre-exercise and 36 post-exercise CBF estimates. Subtraction images were then coregistered to the anatomical image and smoothed using a 6mm full-width, half maximum Gaussian window [37]. CBF area under the curve (AUC), our primary outcome measure, was calculated as the sum of the mean CBF estimate in each region of interest over the duration of acquisition (mL*100g tissue$^{-1}$).

## Blood specimen processing

Immediately after each blood collection timepoint, plasma was centrifuged at 1500 relative centrifugal field (g) (2800 RPM) at 4˚C for 10 minutes. Platelet-rich plasma was then centrifuged in four, 1.5mL aliquots at 1700g (4500 RPM) at 4˚C for 15 minutes. The resulting platelet-poor plasma was separated from the pellet and snap frozen in liquid nitrogen until stored at -80˚C at the end of the visit. Concentrations of IGF-1 (Alpco Diagnostics), VEGF (R&D Systems), and BNDF (R&D systems) were measured in plasma using enzyme linked immunosorbent assays. We then computed a change score between pre-exercise and immediate post-exercise levels for each analyte.

Whole blood was drawn and stored frozen at -80˚C prior to genetic analyses using a Taqman single nucleotide polymorphism (SNP) allelic discrimination assay (ThermoFisher) to determine APOE genotype. Taqman probes were used to determine APOE4, APOE3, and APOE2 alleles to the two APOE-defining SNPs, rs429358 (C_3084793_20) and rs7412 (C_904973_10). Individuals were classified as APOE4 carrier in the presence of 1 or 2 APOE4 alleles (e.g. E3/E4, E4/E4), and remaining individuals were grouped as non-carriers. Blood specimen processing was performed by a trained phlebotomist. Subsequent analyses were performed by trained staff, overseen by an experienced investigator (JKM).

## Statistical analyses

Demographic and intervention differences between APOE4 carriage groups were explored with Welch Two Sample t-test or Fisher's Exact Test as appropriate. Our a priori planned analysis of the primary CBF outcome measure was an independent t-test comparison of CBF AUC between APOE4 carriage groups. Our secondary outcomes were tested via independent t-test comparison of change in blood-based neurotrophic marker levels (post-exercise minus pre-exercise concentration) between e4carriage groups. We also tested an exploratory linear mixed effects model with a random intercept coefficient for each participant and a covariance structure of compound symmetry. P-values were obtained by likelihood ratio tests of the full model against the model without the interaction or factor in question. For our exploratory analyses, we compared change in the AUC of the 2 pre-exercise ASL sequences and the AUC of the final 2 post-exercise ASL sequences across 3 regions of interest (cortex, cerebellum, hippocampus). Age and gender were explored as influential covariates.

Data were captured using REDCap [9]. The analyses for this project were performed using R (base and lme4 packages) [43, 44]. For all analyses p-values less than or equal to 0.05 were considered statistically significant.

# Results

## Participants

A total of 112 individuals were assessed for study eligibility from October 2019 through October 2021. Reasons for exclusion are presented in Fig 1. Enrollment was expanded to 62 in August '21 to increase representation of men and individuals identifying with a racial or ethnic minoritized community. Genotyping of one individual failed, and this person was excluded from analysis. One person withdrew during exercise due to an adverse event, one refused post-exercise MRI, and post-exercise blood collection failed on 2 participants, leaving sample sizes of 59 and 58 for primary and secondary outcomes, respectively.

Self-reported racial and Hispanic/Latino ethnic identity of enrollees was recorded in compliance with National Institute of Health guidance, and approximately reflected the diversity of older adults in the Kansas City region in the 2020 census. We also identified rural residence [45] and calculated the Area Deprivation Index, a geospatial socio-economic disadvantage metric, related to health and dementia risk, to enrich characterization of our participants [46]. We found no evidence of significant differences between carriers and non-carriers in standard demographic measures (p> = 0.3, Table 1).

## Primary outcome

In our pre-specified analysis, we found no evidence of an effect of APOE4 carriage on cortical gray matter post-exercise CBF AUC, see Table 2 (t = 1.3, p = 0.19, 95%CI [-53.9 256.1]). Fig 2 shows CBF AUC for our pre-specified whole gray matter cortical CBF AUC and cerebellar reference region.

## Secondary outcomes

Change in our blood-based markers from pre- to post-exercise, were not significant in any of the neurotrophic factors we explored (Table 2). Pre- to Post-exercise change in VEGF and

**Table 1. Demographics.**

| Characteristic | Overall, N = 61[1] | Non-carrier, N = 38[1] | APOE4 Carrier, N = 23[1] | p-value[2] |
|---|---|---|---|---|
| **Age** | 72.8 (5.2) | 73.3 (5.2) | 72.1 (5.1) | 0.4 |
| **Gender** | | | | 0.3 |
| Men | 20 (33%) | 10 (26%) | 10 (43%) | |
| Women | 41 (67%) | 28 (74%) | 13 (57%) | |
| Non-Binary | 0 (0%) | 0 (0%) | 0 (0%) | |
| **Race** | | | | >0.9 |
| Asian | 1 (1.6%) | 1 (2.6%) | 0 (0%) | |
| Black or African American | 6 (9.8%) | 4 (11%) | 2 (8.7%) | |
| White | 54 (89%) | 33 (87%) | 21 (91%) | |
| **Ethnicity** | | | | >0.9 |
| Non-Hispanic or Latino | 59 (97%) | 37 (97%) | 22 (96%) | |
| Hispanic or Latino | 2 (3.3%) | 1 (2.6%) | 1 (4.3%) | |
| **Rural Residence** | | | | >0.9 |
| Sub/Urban Resident | 58 (97%) | 37 (97%) | 21 (95%) | |
| Rural Resident | 2 (3.3%) | 1 (2.6%) | 1 (4.5%) | |
| **Formal Education** (yrs) | 18.8 (2.8) | 18.8 (2.4) | 18.7 (3.3) | 0.9 |
| **Area Deprivation Index** (National %) | 35.0 (2.0–96.0) | 34.5 (7.0–91.0) | 36.0 (2.0–96.0) | 0.8 |

[1]Mean (standard deviation); n (%); Median (minimum-maximum)

[2]Welch Two Sample t-test; Fisher's exact test

**Table 2. Pre-specified primary and secondary outcomes.**

|  | Overall, N = 59 | Non-carrier, N = 38[1] | APOE4 Carrier, N = 21 |
|---|---|---|---|
| **Whole Gray Matter CBF AUC** | 1,503.6 (259.0) | 1,539.6 (225.5) | 1,438.5 (305.9) |
| **Change in BDNF (pg/mL)** | 110.1 (616.5) | 207.8 (705.9) | -65.6 (362.9) |
| **Change in IGF1 (pg/mL)** | 4.5 (26.0) | 5.7 (27.8) | 2.3 (22.8) |
| **Change in VEGF (pg/mL)** | 0.9 (10.9) | 1.8 (11.1) | -1.0 (10.6) |

Area under cerebral blood flow curve (CBF AUC). Pre to post-exercise change in brain derived neurotrophic factor (BDNF), insulin-like Growth Factor 1 (IGF1) and vascular endothelial growth factor (VEGF). All values are presented as mean (standard deviation).

IGF1 change did not approach significance (p>0.34). Change in BDNF post-exercise was increased but did not reach significance (p = 0.06).

## Exploratory analyses

In our exploratory analyses, we first modeled a 3-way interaction of gray matter region (whole cortical, hippocampus, cerebellum), relative CBF AUC from baseline to post-exercise, and

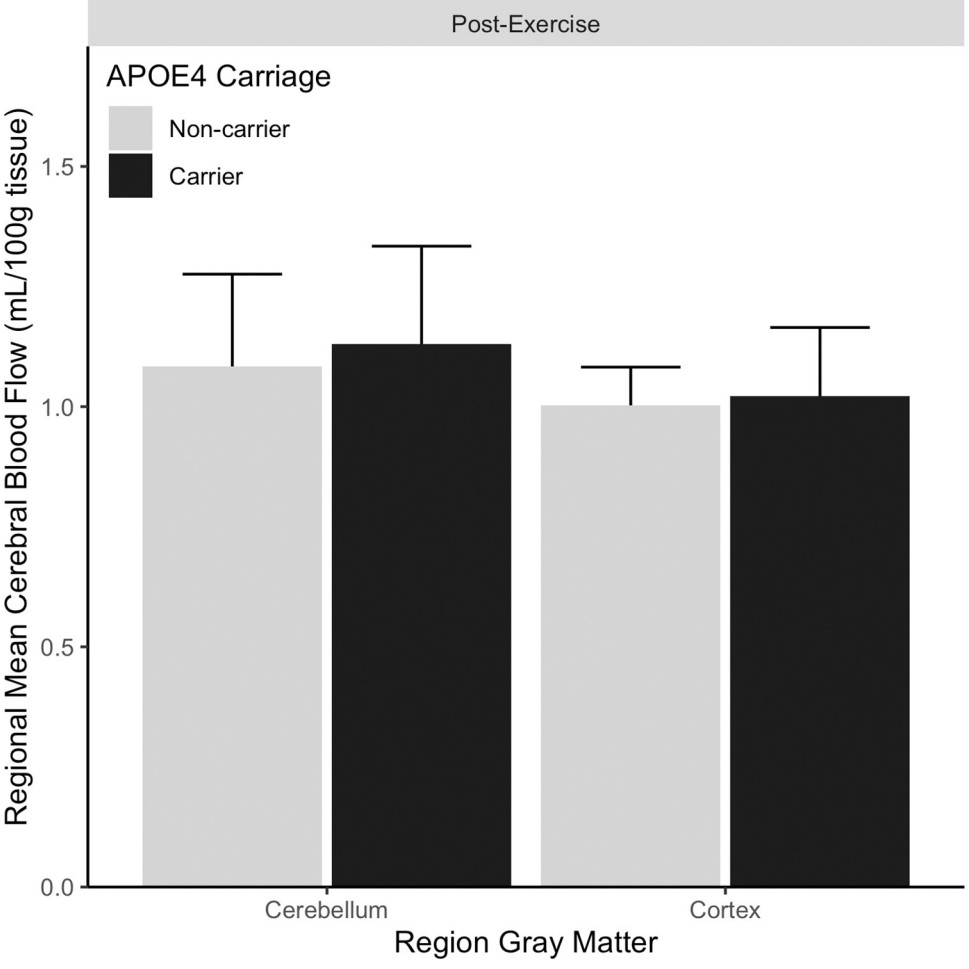

**Fig 2. Cerebral blood flow area under the curve does not differ after exercise based on APOE4 carriage.** Total cerebral blood flow following exercise is plotted for both the primary region of interest, cortical gray matter, and the cerebellar gray matter reference region. Black bars denote APOE4 carriers. Gray bars denote APOE4 non-carriers. Error bars are standard deviation.

APOE4 carriage. CBF was calculated relative to the second, most stable, pre-exercise ASL acquisition. Gender, mean arterial pressure and age were included as a covariate based on preliminary modelling of influential demographic factors. Including the 3-way interaction significantly improved the model fit compared to the reduced model without the interaction of region, APOE4 carriage and CBF AUC ($X^2$ = 15.1, p = 0.03). The presence of the significant 3-way interaction allowed us to perform post-hoc modeling on each region separately. We found that APOE4 carriers had higher post-exercise CBF AUC in the hippocampus ($X^2$ = 3.8, p = 0.05), but not in the whole cortical gray matter ($X^2$ = 2.3, p = 0.12), and not the cerebellum ($X^2$ = 0.89, p = 0.35; S1 Fig). Across all regions, women had significantly higher CBF (p<0.001).

### Adherence and safety

There was 1 adverse event, nausea, during exercise which resulted in termination of the visit and withdrawal of the participant (APOE4 carrier). One person elected not to complete the MRI post exercise (APOE4 carrier). All remaining participants were able to exercise within their identified target heart rate zone. There were no differences in the total power output in Watts, of the exercisers (p = 0.38). APOE4 carriers had a mean power output of 741 (standard deviation 304) and non-carriers had a mean power output of 664 (standard deviation 347).

## Discussion

This is the first study to specifically assess cerebral blood flow (CBF) responses to exercise, comparing those with and without a common genetic Alzheimer's risk factor, APOE4. Our pre-specified analyses found no differences in whole brain CBF post-exercise between APOE4 carriers and non-carriers. Likewise, changes in circulating neurotrophic factor levels immediately post-exercise were not different between carrier and noncarriers. The exploratory experimental approach of this study was designed to investigate the acute physiologic response to exercise, and not investigating exercise as a therapeutic intervention. As such, we explored the regional-specific changes in CBF in the hippocampus, given its differential benefit to exercise interventions and salience in cognitive change and dementia. In our exploratory analyses we found that APOE4 carriers display a greater increase in hippocampal region CBF in the acute response following exercise that were not present in whole brain or the cerebellar region, the latter serving as a reference region. These findings extend prior work showing similar hyperemic response in the hippocampus [47], and provide initial evidence that APOE4 carriers demonstrate greater hyperemia within the hippocampus than their non-carrier peers immediately after an acute exercise bout. Further, the heterogeneity of immediate post-exercise neurotrophin response across all older adults in the present study identify an area of future exploration for future research investigating acute physiologic responses to aerobic exercise. These findings provide an individualized framework for acute physiologic responses to an acute bout of aerobic exercise. Our results support a precision-medicine approach for the characterization and targeting of physiologic substrates with exercise interventions to benefit brain health.

### Effect of APOE4 genotype on acute exercise-induced cerebral blood flow

Prior reports of hippocampal blood flow change in acute response to exercise have been inconsistent, with both increases and decreases reported [47–50]. Our findings are consistent with prior work demonstrating chronically increased cerebral blood flow in the hippocampi of young adults following an exercise intervention [51], and further highlight the APOE4 genotype-by-hippocampal interaction effect that should be considered in aging populations.

Though the present study is one of the first investigations of the immediate acute effects of aerobic exercise in older adults, Alfini et al reported that short periods (i.e. 10 days) of sedentary behavior have a powerful reverse effect for reducing hippocampal CBF in highly active older adults [52]. The present results build up on these previous findings, together suggesting that hippocampal brain structures in older adults of this known risk allele are highly sensitive to changes in physical activity behaviors. Importantly, our findings provide a foundation for an individualized framework and brain region-specific analyses when studying the effects of exercise on cerebral blood flow. This may be a critical next step for linking cognitive maintenance to exercise effects, as prior work has failed to demonstrate a direct relationship between proxies of cerebral blood flow (transcranial Doppler) and cognition [53].

### Neurotrophin factors show no change immediately following acute aerobic exercise in older adults

In contrast to previous reports in neurotypical young adults, we observed no exercise-induced change in blood neurotrophin concentration in older adults in the present study, regardless of APOE4 carrier status. This finding was surprising given that previous studies in younger adults report robust increases in these neurotrophic factors, IGF1, VEGF, and BDNF among others [20, 54–57]. Exercise-induced increases in neurotrophic factors have been associated with neurogenesis and angiogenesis in rodent models and are thought to explain brain health and cognitive benefits of exercise interventions [51, 58, 59]. However, in almost all cases, CBF and neurotrophins in human studies have been measured following an extended period of rest, without a challenging stimulus. Given that benefits would necessarily result from discrete, repeated exposures to an exercise intervention, measuring during inactivity potentially obscures important dynamic adaptations or capacities. Indeed, a challenging stimulus such as acute bout of aerobic exercise may be necessary to sufficiently study local and systemic effects on the brain [60, 61]. Methodological differences may also contribute to these differences. We chose to focus on platelet poor plasma, as neurotrophins are released from platelets following freeze-thaw cycles [62]. We believed this approach would give a more accurate representation on circulating, rather than stored, biomarker concentrations. Future work should consider that these biomarkers may have a delayed increase after exercise stimulus onset. Because APOE4 has been shown to influence release of BDNF and interact with VEGF, additional investigation is warranted [63–65]. Further, given there was no increase in neurotrophins between groups immediately following exercise, the greater change in hippocampal CBF immediately following exercise in older adult APOE4 carriers thus appears to be mechanistically driven by different factors than that observed in younger adults. Future investigations may test whether other physiologic factors (e.g. blood lactate) that may drive cerebral perfusion responses, and may further explain the specificity of such responses in hippocampal brain regions.

### Limitations

This study has several limitations. First, we did not identify CBF change in our pre-specified primary outcome. At the time of inception, National Institute of Health guidance classified all exercise experimental designs as clinical trials. Following CONSORT guidance, we declared a priori outcomes of interest despite relative uncertainty in how to quantify our time-course data. But the effect size of APOE4-related differences in our pre-specified primary outcome was insufficient to reject the null hypothesis. Thus, we feel justified in presenting our alternative analysis. Second, our groups are unbalanced. Though we made significant efforts to over-represent APOE4 carriers [35], our final sample approximates the distribution of the E4 in the

US population. Given the advantage of high spatial resolution and sensitivity to cerebral perfusion changes, the present study utilized a MR imaging method to quantify cerebral blood flow. This method limits our ability to interpret CBF changes during the exercise bout that may have influenced immediate post-exercise CBF changes. Though this MR imaging method is currently regarded as the most accurate and precise method to quantify cerebral blood flow, the CBF measurements may be sensitive to factors such as day-to-day variability and circadian cycle [66]. As we have previously reported, the recovery time course for CBF appears to be relatively independent of blood pressure changes [35]. However, future work should emphasize accurate measurement of blood pressure, respiratory rate, and heart rate during the exercise bout to test the effect of blood pressure changes on CBF. Given the strong link between cardiovascular health and cognition with aging, our exclusion of older adults with cognitive impairment or dementia could have biased our sample towards individuals with higher vascular health than the typical older adult population. Finally, though our sample demonstrates racial and socioeconomic diversity, both inclusion criteria that limited the enrollment of individuals with severe cardiovascular disease and the underrepresentation of other racial identities, men, and additional sources of diversity limit the broad generalizability of this work.

## Conclusion

We conducted the first comparison of the effect of a common Alzheimer's risk gene, APOE4, on post-exercise cerebral blood flow and common neurotrophic changes following moderate intensity aerobic exercise. Our method of characterizing cerebral blood flow recovery may provide new avenues for MRI quantification of perfusion change. By using this method, we extended prior work showing that the hippocampus experiences great post-exercise blood flow increases in older adult APOE4 carriers. Investigation of the key mechanisms by which aerobic exercise supports cognition and brain health will continue to have important implications for future work by optimizing prescribed exercise interventions and specifying appropriate outcomes of interest.

## Supporting information

**S1 Checklist. CONSORT 2010 checklist of information to include when reporting a randomized trial**[*]**.**
(PDF)

**S1 Fig. Regional blood flow pre- and post-exercise.** The figure shows relative cerebral blood flow in three regions of interest. Pre- and post-exercise time frames are equivalent, ~12 minutes of arterial spin labeling data collection. The hippocampus demonstrated an increase in post-exercise cerebral blood flow over pre-exercise in APOE4 carriers only (p = 0.05). The white bar is pre-exercise for APOE4 non-carriers. The light gray bar is post-exercise for APOE4 non-carriers. The dark graybar is pre-exercise for APOE4 carriers. The black bar is post-exercise for APOE4 carriers. Error bars are standard deviation. Cerebral blood flow is shown in percentage of the second PCASL acquisition before exercise.
(TIF)

**S1 Protocol. Dynamic arterial measurement in cerebrum (DYNAMIC).**
(PDF)

## Author Contributions

**Conceptualization:** Eric D. Vidoni, Jill K. Morris, Jacqueline A. Palmer, Rebecca J. Lepping, Phil Lee, Jonathan D. Mahnken, Laura E. Martin, Sandra A. Billinger.

**Data curation:** Eric D. Vidoni, Jill K. Morris, Jacqueline A. Palmer, Yanming Li, Dreu White, Casey S. John.

**Formal analysis:** Jacqueline A. Palmer, Yanming Li, Jonathan D. Mahnken.

**Funding acquisition:** Jill K. Morris.

**Investigation:** Eric D. Vidoni, Jill K. Morris, Dreu White, Paul J. Kueck, Casey S. John, Robyn A. Honea.

**Methodology:** Eric D. Vidoni, Dreu White, Paul J. Kueck, Casey S. John, Robyn A. Honea, Phil Lee, Jonathan D. Mahnken, Laura E. Martin, Sandra A. Billinger.

**Project administration:** Eric D. Vidoni, Jill K. Morris, Dreu White, Paul J. Kueck, Rebecca J. Lepping, Jonathan D. Mahnken, Sandra A. Billinger.

**Resources:** Eric D. Vidoni, Jill K. Morris.

**Software:** Eric D. Vidoni, Robyn A. Honea, Rebecca J. Lepping, Phil Lee.

**Supervision:** Eric D. Vidoni, Jonathan D. Mahnken, Sandra A. Billinger.

**Validation:** Eric D. Vidoni.

**Visualization:** Eric D. Vidoni, Jill K. Morris.

**Writing – original draft:** Eric D. Vidoni, Jill K. Morris, Jacqueline A. Palmer.

**Writing – review & editing:** Eric D. Vidoni, Jill K. Morris, Jacqueline A. Palmer, Yanming Li, Dreu White, Paul J. Kueck, Casey S. John, Robyn A. Honea, Rebecca J. Lepping, Phil Lee, Jonathan D. Mahnken, Laura E. Martin, Sandra A. Billinger.

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
