## [Decision Letter · Decision Letter 0]

5 May 2022

PONE-D-22-06875Dementia Risk and Dynamic Response to Exercise: A non-randomized clinical trialPLOS ONE

Dear Dr. Vidoni,

Thank you for submitting your manuscript to PLOS ONE. After careful consideration, we feel that it has merit but does not fully meet PLOS ONE’s publication criteria as it currently stands. Therefore, we invite you to submit a revised version of the manuscript that addresses the points raised during the review process.

We look forward to receiving your revised manuscript.

Kind regards,

Walid Kamal Abdelbasset, Ph.D.

Academic Editor

PLOS ONE

Journal Requirements:

"This study was funded by grants from the national institutes of health R21 AG061548, P30 AG072973 and P30 AG035982, and the Leo and Anne Albert Charitable Trust. The Hoglund Biomedical Imaging Center is supported by a generous gift from Forrest and Sally Hoglund and funding from the National Institutes of Health including S10 RR29577, and UL1 TR002366."

We note that you have provided funding information that is not currently declared in your Funding Statement. However, funding information should not appear in the Funding section or other areas of your manuscript. We will only publish funding information present in the Funding Statement section of the online submission form. 

"This study was funded by grants from the national institutes of health R21061548 (EDV), P30 AG072973 (EDV, JKM, RAH, RJL, JM, SAB), and the Leo and Anne Albert Charitable Trust. (EDV)."

Reviewers' comments:

Reviewer's Responses to Questions

**Comments to the Author**

1. Is the manuscript technically sound, and do the data support the conclusions?

Reviewer #1: Yes

Reviewer #2: Partly

2. Has the statistical analysis been performed appropriately and rigorously? 

Reviewer #1: Yes

Reviewer #2: Yes

3. Have the authors made all data underlying the findings in their manuscript fully available?

Reviewer #1: Yes

Reviewer #2: Yes

4. Is the manuscript presented in an intelligible fashion and written in standard English?

Reviewer #1: Yes

Reviewer #2: Yes

5. Review Comments to the Author

Reviewer #1: A non-randomized clinical trial was conducted which aimed to examine the effect of aerobic exercise on brain blood flow and blood neurotrophic factors in older adults with and without the APOE4 allele. Cerebral blood flow AUC increased immediately after exercising regardless of APOE4 allele status. No significant changes were observed in the neurotrophic factors for either group immediately following exercise.

Minor revisions:

1- Abstract: In addition to the count, include the percentage female.

2- Line 193: Clarify if the t-tests for comparing pre- to post-exercise change were paired t-test.

3- Line 195: State the underlying covariance structure used in the linear mixed effects model and the criteria for selecting it.

4- Specify the level of significance. For instance, add the following statement, filling in the value for x.xx. P-values less than x.xx were considered statistically significant.

5- Use consistent notation for standard deviation. The standard notation is SD. Spell out the abbreviation at its first use.

Reviewer #2: Dear authors in term of novelty this is a new study but kindly see the attached file and see the required modifications in the comments and try to do it as mentioned for the soundness of your paper to be published

6. PLOS authors have the option to publish the peer review history of their article (what does this mean?). If published, this will include your full peer review and any attached files.

Reviewer #1: No

Reviewer #2: No

---

## [Author Response · Author response to Decision Letter 0]

3 Jun 2022

We appreciate the timely and thoughtful review of our manuscript, Dementia Risk and Dynamic Response to Exercise: A Non-randomized Clinical Trial, provided by the reviewers. We have carefully considered each critique and provided our response below. To assist reviewers, in most cases we have provided the critique numbered, our response in plain text, and the revision to the manuscript “quoted”. 

Reviewer #1

Minor revisions:

1- Abstract: In addition to the count, include the percentage female.

We have added the percentage female to the abstract

“Methods: Sixty-two older adults (73±5 years old, 41 female [67%])…”

2- Line 193: Clarify if the t-tests for comparing pre- to post-exercise change were paired t-test.

We apologize for the lack of clarity. We performed independent t-tests to compare APOE4 carrier and non-carrier groups. The measures of interest were 1) the post-exercise area under the curve, and 2) the pre-to-post exercise change [change measure] in neurotrophin concentration. We have revised our language for improved clarity.

“Our a priori planned analysis of the primary CBF outcome measure was an independent t-test comparison of CBF AUC between APOE4 carriage groups, assuming unequal variance. Our secondary outcomes were tested via independent t-test comparison of change in blood-based neurotrophic marker levels (post-exercise minus pre-exercise concentration) between e4carriage groups.”

3- Line 195: State the underlying covariance structure used in the linear mixed effects model and the criteria for selecting it.

Since we fit a linear mixed effects model with a random intercept, the resulting covariance structure for our observations is compound symmetry. We have stated this in the text at the specified location.

4- Specify the level of significance. For instance, add the following statement, filling in the value for x.xx. P-values less than x.xx were considered statistically significant.

Thank you for identifying this oversight. We have clarified our a priori defined alpha.

“For all analyses we set alpha = 0.05.”

5- Use consistent notation for standard deviation. The standard notation is SD. Spell out the abbreviation at its first use.

We have corrected the use of “SD” to “standard deviation”. No abbreviation is necessary in the manuscript as it is only a handfull of times in the footnotes of tables or text.

Reviewer #2

1. Choose a more descriptive short title.

We have revised the Short Title.

“Short Title: Cerebral blood flow response to exercise”

2. Apolipoprotein E, first illustrate the abbreviation then mention it.

This has been corrected.

3. The results of previous study should not be mentioned in abstract section it should be in the discussion

Reference to prior studies have been removed from the abstract Conclusions section

“Results from this study provide an initial characterization of the acute brain blood flow and neurotrophin responses to a bout of exercise in older adults with and without this known risk allele for cardiovascular disease and Alzheimer’s disease.”

4. Where is the key words

Keywords were entered into the PLOS One submission management system. We have added them to the manuscript.

“Keywords: cerebral blood flow; aerobic exercise; Alzheimer's disease; perfusion”

5. Explain the rationale of the study. Please delete information unrelated to objective so that the section is short and sweet. Kindly focus on three elements of introduction.

a. What is known about the topic? (Background)

b. What is not known? (The research problem)

c. Why the study was done? (Justification)

Objective is not clear as mentioned above.

Thank you for the opportunity to clarify the rationale and objective of the present study. We have edited the Introduction section to improve the clarity and more concisely motivate our primary objectives. Of particular note, we have clarified our statement on the scientific knowledge gap and study premise. 

“… whether older individuals who possess APOE4 show differential CBF responses to an acute bout of exercise has not been investigated. Understanding acute physiologic responses to is important as any benefits of exercise will necessarily result from the cumulative effects of these brief regular exercise exposures.

To further characterize potential intermediary mechanisms between exercise and brain health we designed the present study to assess the immediate CBF response to a single bout of acute exercise.”

6. Why did you choose MRI explain ? and add references to all measures

Thank you for prompting this addition. In addition to our detailed description and appropriate references for the PCASL analyses procedure, we have added the following statement and reference. 

“Arterial spin labeling via magnetic resonance imaging was chosen to capture CBF due to its advantage in spatial localization and ability to yield a physiologically quantifiable outcome.(Borgovac et al, 2012)”

Since the time of original submission, we have come across new methodologies for image processing that may improve the quality of signal processing. To improve the rigor of this work, we have implemented these new analyses approaches in this manuscript revision, including the use of smoothing of the PCASL signal, and have updated citations, and CBF measures reported in the results and Table 2. The outcomes of our statistical analyses remained similar after implementation of the revised CBF analyses methodology and our interpretation of the findings is unchanged.

7. Mention who did blood specimen processing

We have added reference to who performed the blood specimen processing.

“Blood specimen processing was performed by a trained phlebotomist. Subsequent analyses were performed by trained staff, overseen by an experienced co-investigator (JKM).”

8. Results need to follow ABC (accuracy, brevity, clarity)

Kindly frame it along the following elements of results

i. Text to tell the story

ii. Tables to summarize the evidence

iii. Figures to highlight the main findings

In compliance with the study sponsors at the National Institutes of Health (NIH), we have followed standard NIH Consolidated Standards of Reporting of Trials (CONSORT) guidelines for reporting clinical trial results: http://www.consort-statement.org/ . 

 . We have specified primary and secondary outcomes, provided tables to summarize the main evidence, andfigures to highlight the main findings.

9.This part [enrollment] should be placed in the method section not the results

We appreciate the perspective on the optimal location for the enrollment narrative. Throughout the manuscript we have followed standard CONSORT reporting. CONSORT defines the enrollment narrative as part of the Results (Items 13a/b and 14a/b on the CONSORT checklist).

10. Why didn’t you do intention to treat analysis for this drop out

We appreciate this concern, and after further consideration we feel that as only 3 people were missing for the primary outcome the number of remaining subjects included in our analysis is sufficient for estimation and statistical inference. Thus, we opted for this approach as opposed to risk inducing potential biases through imputation. 

11. State the underlying covariance structure used in the linear mixed effects model and the criteria for selecting it.

Since we fit a linear mixed effects model with a random intercept, the resulting covariance structure for our observations is compound symmetry.

We have stated this in the methods section: 

“We also tested an exploratory linear mixed effects model with a random intercept coefficient for each participant and a covariance structure of compound symmetry.”

12. Describe sources of potential bias and imprecision.

We agree that further elaboration on sources of potential bias and imprecision would improve the readers’ ability to carefully evaluate the results of the present study. We have edited the Discussion section to include the following limitations: 

“…Given the advantage of high spatial resolution and sensitivity to cerebral perfusion changes, the present study utilized a MR imaging method to quantify cerebral blood flow. This method limits our ability to interpret CBF changes during the exercise bout that may have influenced immediate post-exercise CBF changes. Though this MR imaging method is currently regarded as the most accurate and precise method to quantify cerebral blood flow, the CBF measurements may be sensitive to factors such as day-to-day variability and circadian cycle (Heijtel et al 2014).”

“…Given the strong link between cardiovascular health and cognition with aging, our exclusion of older adults with cognitive impairment or dementia could have biased our sample towards individuals with higher vascular health than the typical older adult population.”

13. Generalizability of the trial findings need to be put.

We have included the following statement in the Limitations.

“Finally, though our sample demonstrates racial and socioeconomic diversity, both inclusion criteria that limited the enrollment of individuals with severe cardiovascular disease and the underrepresentation of other racial identities, men, and additional sources of diversity limit the broad generalizability of this work.”

---

## [Decision Letter · Decision Letter 1]

10 Jun 2022

PONE-D-22-06875R1Dementia Risk and Dynamic Response to Exercise: A non-randomized clinical trialPLOS ONE

Dear Dr. Vidoni,

Thank you for submitting your manuscript to PLOS ONE. After careful consideration, we feel that it has merit but does not fully meet PLOS ONE’s publication criteria as it currently stands. Therefore, we invite you to submit a revised version of the manuscript that addresses the points raised during the review process.

We look forward to receiving your revised manuscript.

Kind regards,

Walid Kamal Abdelbasset, Ph.D.

Academic Editor

PLOS ONE

Journal Requirements:

Reviewers' comments:

Reviewer's Responses to Questions

**Comments to the Author**

1. If the authors have adequately addressed your comments raised in a previous round of review and you feel that this manuscript is now acceptable for publication, you may indicate that here to bypass the “Comments to the Author” section, enter your conflict of interest statement in the “Confidential to Editor” section, and submit your "Accept" recommendation.

Reviewer #1: (No Response)

Reviewer #2: All comments have been addressed

2. Is the manuscript technically sound, and do the data support the conclusions?

Reviewer #1: Yes

Reviewer #2: Yes

3. Has the statistical analysis been performed appropriately and rigorously? 

Reviewer #1: Yes

Reviewer #2: Yes

4. Have the authors made all data underlying the findings in their manuscript fully available?

Reviewer #1: Yes

Reviewer #2: Yes

5. Is the manuscript presented in an intelligible fashion and written in standard English?

Reviewer #1: Yes

Reviewer #2: Yes

6. Review Comments to the Author

Reviewer #1: Minor revision:

The following statement is vague since the denotation of the lower case a is nonstandard. "For all analyses we set a= 0.05." Consider writing, "P-values less than 0.05 were considered statistically significant."

Reviewer #2: Thanks for submitting all required modifications and response to all required illustrations on this paper

7. PLOS authors have the option to publish the peer review history of their article (what does this mean?). If published, this will include your full peer review and any attached files.

Reviewer #1: No

Reviewer #2: **Yes: **Marwa Eid

---

## [Author Response · Author response to Decision Letter 1]

10 Jun 2022

We appreciate the timely review of our first revision of our manuscript, Dementia Risk and Dynamic Response to Exercise: A Non-randomized Clinical Trial, provided by the reviewers. We have performed the final requested changes.

Reviewer #1: Minor revision:

The following statement is vague since the denotation of the lower case a is nonstandard. "For all analyses we set a= 0.05." Consider writing, "P-values less than 0.05 were considered statistically significant."

We have changed the statement in the manuscript as requested.

---

## [Decision Letter · Decision Letter 2]

23 Jun 2022

Dementia Risk and Dynamic Response to Exercise: A non-randomized clinical trial

PONE-D-22-06875R2

Dear Dr. Vidoni,

We’re pleased to inform you that your manuscript has been judged scientifically suitable for publication and will be formally accepted for publication once it meets all outstanding technical requirements.

Kind regards,

Walid Kamal Abdelbasset, Ph.D.

Academic Editor

PLOS ONE

Additional Editor Comments (optional):

Reviewers' comments:

Reviewer's Responses to Questions

**Comments to the Author**

1. If the authors have adequately addressed your comments raised in a previous round of review and you feel that this manuscript is now acceptable for publication, you may indicate that here to bypass the “Comments to the Author” section, enter your conflict of interest statement in the “Confidential to Editor” section, and submit your "Accept" recommendation.

Reviewer #1: All comments have been addressed

2. Is the manuscript technically sound, and do the data support the conclusions?

Reviewer #1: (No Response)

3. Has the statistical analysis been performed appropriately and rigorously? 

Reviewer #1: (No Response)

4. Have the authors made all data underlying the findings in their manuscript fully available?

Reviewer #1: (No Response)

5. Is the manuscript presented in an intelligible fashion and written in standard English?

Reviewer #1: (No Response)

6. Review Comments to the Author

Reviewer #1: (No Response)

7. PLOS authors have the option to publish the peer review history of their article (what does this mean?). If published, this will include your full peer review and any attached files.

Reviewer #1: No

---

## [Editor Report · Acceptance letter]

29 Jun 2022

PONE-D-22-06875R2 

Dementia Risk and Dynamic Response to Exercise: A Non-randomized Clinical Trial 

Dear Dr. Vidoni:

I'm pleased to inform you that your manuscript has been deemed suitable for publication in PLOS ONE. Congratulations! Your manuscript is now with our production department. 

Kind regards, 

on behalf of

Dr. Walid Kamal Abdelbasset 

Academic Editor

PLOS ONE